# The Effects of Nanosilica on Mechanical Properties and Fracture Toughness of Geopolymer Cement

**DOI:** 10.3390/polym13132178

**Published:** 2021-06-30

**Authors:** Cut Rahmawati, Sri Aprilia, Taufiq Saidi, Teuku Budi Aulia, Agung Efriyo Hadi

**Affiliations:** 1Doctoral Program, School of Engineering, Universitas Syiah Kuala, Banda Aceh 23111, Indonesia; cut_19@mhs.unsyiah.ac.id (C.R.); taufiq_saidi@unsyiah.ac.id (T.S.); aulia@unsyiah.ac.id (T.B.A.); 2Department of Civil Engineering, Engineering Faculty, Universitas Abulyatama, Aceh Besar 23372, Indonesia; 3Department of Chemical Engineering, Engineering Faculty, Universitas Syiah Kuala, Banda Aceh 23111, Indonesia; 4Department of Civil Engineering, Engineering Faculty, Universitas Syiah Kuala, Banda Aceh 23111, Indonesia; 5Department of Menchanical Engineering, Universitas Malahayati, Lampung 35153, Indonesia; efriyo@malahayati.ac.id

**Keywords:** geopolymer, epoxy, fracture toughness, tensile strength, flexural strength

## Abstract

Nanosilica produced from physically-processed white rice husk ash agricultural waste can be incorporated into geopolymer cement-based materials to improve the mechanical and micro performance. This study aimed to investigate the effect of natural nanosilica on the mechanical properties and microstructure of geopolymer cement. It examined the mechanical behavior of geopolymer paste reinforced with 2, 3, and 4 wt% nanosilica. The tests of compressive strength, direct tensile strength, three bending tests, Scanning Electron Microscope-Energy Dispersive X-ray (SEM/EDX), X-ray Diffraction (XRD), and Fourier-transform Infrared Spectroscopy (FTIR) were undertaken to evaluate the effect of nanosilica addition to the geopolymer paste. The addition of 2 wt% nanosilica in the geopolymer paste increased the compressive strength by 22%, flexural strength by 82%, and fracture toughness by 82% but decreased the direct tensile strength by 31%. The microstructure analysis using SEM, XRD, and FTIR showed the formation of calcium alumina-silicate hydrate (C–A–S–H) gel. The SEM images also revealed a compact and cohesive geopolymer matrix, indicating that the mechanical properties of geopolymers with 2 wt% nanosilica were improved. Thus, it is feasible for nanosilica to be used as a binder.

## 1. Introduction

Over the last few years, the cement industry has caused an effect on global warming with carbon dioxide emissions, estimated to be nearly 5–7% of total CO_2_ emissions in the environment [1]. Therefore, in recent years, most of the research has focused on geopolymer materials that can be synthesized from an alkaline activation process of various organic materials or low-cost industrial byproducts, such as fly ash, rice husk ash, furnace slag as green materials. Fly ash is widely used as a raw material in composite geopolymer [2,3]. Alkaline solution will react silica and alumina in fly ash through a polymerization process and produce a sodium–aluminosilicate gel.

Nanosilica is the most widely-used nanomaterial in both common cement and geopolymers to improve cement properties due to its reactive pozzolans and pore-filling effect [4,5]. It consists of silicon dioxide (SiO_2_) in crystalline and amorphous forms. Amorphous nanosilica is the most commonly used type of nanoconcrete [6]. Nanosilica particles vary in size, ranging between 5–658 nm for various types of silica-based nanomaterial products [7,8].

The addition of nanosilica not only accelerates the rate of polymerization reaction but also promotes the development of calcium silicate hydrates (C-S-H) and sodium aluminosilicate hydrates (N–A–S–H) in natural pozzolan-based polymers [9]. The best results were obtained when 1 wt% nano-SiO_2_ was added to a metakaolin-based geopolymer with a w/s ratio of 1.03 [10]. The addition of nanosilica to the geopolymer mortar increases the dissolved silica, driving the accelerated rate of geopolymerization by forming long-chain silicate oligomers in the geopolymer matrix [11]. Moreover, the larger surface area of nano-silica particles will accelerate the geopolymerization process [12]. Ibrahim et al. [9] reported that the addition of nanosilica not only accelerates the rate of polymerization reaction but also promotes the development of calcium silicate hydrates (C–S–H) and sodium aluminosilicate hydrates (N–A–S–H) in natural pozzolan-based geopolymer concrete. Adak et al. [13] investigated the performance of nanosilica in fly ash-based geopolymer concrete and observed that in the presence of nanosilica, the dissolution rate of Si and Si-Al increases and accelerates the rate of geopolymerization. This is related to the amorphous nature of nanosilica and its high surface area.

Recycling and reusing agricultural waste byproducts such as rice husk ash has been a concern of researchers in recent years because of the environmental problems it causes. The high content of silica in rice husk ash makes this material an option to be physically converted into a nanomaterial and used as a reinforcement in this study. The use of nanosilica in cement geopolymers faces several obstacles. The present fact that geopolymers with many favorable characteristics have not been as widely used as Portland cement. The widespread use of geopolymers is constrained by several limitations such as high porosity, slow setting, and subsequent slow strength development [11]. These limitations need to be improved to increase the use of geopolymers in concrete as an alternative to the binder.

The effect of nanosilica in Portland cement has been extensively investigated and found to be an effective additive to the development of mechanical and microstructural properties. However, research on the nanosilica effect of rice husk ash on the mechanical properties, microstructure, and fracture toughness of geopolymers has not been broadly discussed in the literature. Therefore, this study aimed to investigate the effect of physically processed nanosilica on geopolymer cement’s mechanical/microstructure properties such as compressive strength, direct tensile strength, flexural strength, ductility, fracture toughness, Scanning Electron Microscope (SEM/EDX), X-ray Diffraction (XRD) and Fourier Transform Infrared Spectroscopy (FTIR). The optimal amount of nanosilica is determined by setting the silica content at 0–4 wt%. The mechanical strength and microstructural development of geopolymer paste samples containing different nanosilica are treated in the curing process at room temperature studied.

## 2. Materials and Methods

### 2.1. Materials

Class C fly ash from the Nagan Raya power plant in Aceh Province, Indonesia, nanosilica from physically-processed rice husk ash, and epoxy resin made in Germany purchased from the Indonesia Chemical Reagent Company, were used in this study. A scanning electron microscope (SEM, Vega3–Tescan, Kohoutovice, Czech Republic) test was used to observe the morphology of fly ash and Transmission Electron Microscope (TEM, JEOL JEM 1400, Tokyo, Japan) for nanosilica shown in Figure 1. Nanosilica was processed according to previous research results [14] by applying a ball mill on white rice husk ash for 10 h at a speed of 600 rpm to change the size of rice husk ash to nano. This process obtained nanosilica with an average particle size of 339.09 nm. The particle size of nanosilica was measured using a Particle Size Analyzer (PSA, Horiba SZ–100V2, Kyoto, Japan). The chemical composition of fly ash and nanosilica is presented in Table 1.

The epoxy resin used was waterborne bisphenol-epoxy resin, and waterborne polyamine epoxy curing agents were obtained from commercial suppliers. The alkaline activator is a mixture of sodium hydroxide solutions and sodium silicate solutions. In this study, a sodium hydroxide solution with 10 M concentration was used, which was made by mixing 97–98% pure palette with water. The ratio of SiO_2_ and Na_2_O mass from sodium silicate solution was 2.61 (SiO_2_ = 24.19%, Na_2_O = 12.81% and water = 63%). Thermophysical properties of material used is shown in Table 2.

### 2.2. Geopolymer Mixes Containing Nanosilica

The composition of the geopolymer paste mixture was made proportionally based on previous work carried out by lab workers at room temperature [15,16]. The ratio for the mixed formula is based on the ratio Na_2_SiO_3_/NaOH = 1. Table 3 shows the mixture proportions used in the paste geopolymer. The mixing process was carried out by first passing fly ash on a 200-mesh sieve, drying at 70 °C for 1 h. After that, we gradually mixed the nanosilica with dosage 2, 3, 4 wt% and stirred for 5 min. Mixing was carried out using a Hobart mixer. After that, we added activator solutions and stirred for 20 min at a stirring speed of 800 rpm as recommended. The epoxy solution used was 20% by weight matrix with a 1:1 ratio of epoxy resin and catalyst. After that, the two ingredients, namely the geopolymer paste and epoxy resin, were stirred again for 5 min at a speed of 1350 rpm, as recommended (Roviello et al., 2016) [17]. After the mixture was evenly mixed, the paste was poured into a 50 × 50 × 50 mm^3^ cube mold according to the ASTM C109 standard [18] and stored at room temperature. After one day of age, the specimens were demolded.

### 2.3. Testing Procedures

#### 2.3.1. Mechanical Properties

Compressive strength testing was carried out according to ASTM C109, 2001 [18] standards. The size of the test object used was a 5 × 5 × 5 cm cube. The calculation was carried out according to the following equation.
(1)$$\sigma_{c}=\frac{P}{A\;}$$
where $\sigma_{c}$ = compressive strength (MPa), P = total load on the specimen at failure (N), and *A* = surface area of the specimen (m^2^). In all mechanica l properties testing, 3 specimens were used in each treatment to get the average value and standard deviation.

The dog bone shape specimens were used to investigate the tensile behavior. The testing machine and typical dog bone shape specimens are shown in Figure 2.

The direct tension test in this study was carried out according to the ASTM C 307-03 standard [19,20] by using Testometric materials testing machines with a loading rate of 30 mm/min.

From this test, the ductility (µ) was calculated based on the following equation.
(2)$$\;µ=\frac{\Delta u}{\Delta y}$$
where Δu is the displacement at ultimate load (Pu = 80% Pmax), and Δy is the displacement at first yield [21].

Three-point bending tests were carried out according to ASTM D5045-14 [19] using specimens with the size of 40 × 40 × 160 mm, and the crack length (a0) was 20 mm. This test was performed using the Testometric materials testing machines with a 10 mm/min loading rate. The test setup schematic is shown in Figure 3.

The flexural strength (σ_F_) was determined according to the formula:(3)$$\sigma_{F}=\frac{3{\;P}_{m}\;S}{2{\;\mathrm{BW}}^{2}}$$
where P_m_ is the maximum load, S is the specimen span, B is the specimen width, and W is the specimen thickness.

This test also determined fracture toughness (K_IC_) by using the following formula:(4)$$K_{\mathrm{IC}}=\frac{P_{m}\;S}{{\mathrm{BW}}^{3/2}}\left(\frac{a}{W}\right)$$
(5)$$\;ƒ\left(\frac{a}{W}\right)=\frac{3{\left(a/W\right)}^{1/2}\left[1.99-\left(a/W\right)\left(1-\;a/W\right)\;\times \;\left(2.15-3.93a/W\;+2.7a^{2\;}/W^{2}\right)\right]}{2\left(1+2a/W\right){\left(1-\;a/W\right)}^{3/2}},$$
where ƒ (a/W) is the polynomial correction factor, (a/W) is the ratio of crack length to the thickness of 0.4.

#### 2.3.2. Characterization of Materials

The morphology of the specimens was evaluated by SEM and for the elements contained therein was also utilized an Energy Dispersive X-ray (EDX) detector (Vega 3-Tescan). XRD analysis was implemented to characterize geopolymer paste specimens with the addition of nanosilica. The XRD testing tool used was the Shimadzu XRD-7000. The X-ray spectrum used was 1017–1020 Hz which has an energy of 103–106 eV. FTIR (Shimadzu-IRPrestige-21) was used to see the functional groups of the geopolymer paste due to the addition of nanosilica. Data are presented in wavenumbers from 4000 to 400 cm^−1^.

## 3. Results

### 3.1. Direct Tensile Strength

Figure 4a,b show the typical stress–strain curves and the maximum tensile strength of the geopolymer cement paste added with nanosilica at the age of 28 days. Figure 4a shows that the addition of 2, 3, and 4 wt% nanosilica did not significantly increase the tensile strength. However, the addition of nanosilica shows a change in the resulting stress-strain curve. When the failure happened, the stress on the sample without the addition of nanosilica immediately dropped to zero. This is different from what happened to the samples that had been added with nanosilica, which showed that there was a drop in stress that did not occur suddenly. The failure of the material due to the addition of nanosilica did not occur suddenly. Figure 4b shows a graph of the maximum stresses resulting from the inclusion of nanosilica 2, 3, and 4 wt%.

The tensile strength due to the addition of 0, 2, 3, and 4 wt% nanosilica were 1.37, 0.95, 0.82, and 0.77 MPa, respectively. Its value obtained in this study was greater than that done with the addition of cornsilk reinforced cement (0.4–0.6 MPa) [22], but it was a small value compared to other studies such as the addition of metakaolin and rice husk ash (0.92–1.91 MPa) [23], steel slag (0.66–1.30 MPa) [20] and polyethylene fiber (3.25–3.43 MPa) [24]. The addition of nanosilica reduced the maximum tensile strength, but nanosilica has not been able to increase the maximum tensile stress. This is because nanosilica only has the ability to fill pores and is unable to withstand the growth of cracks once elastic.

From the stress–strain curve analysis and the maximum tensile strength, it can be concluded that the addition of nanosilica did not increase the tensile strength of the cement paste but actually decreased it. However, the inclusion of 2, 3, 4 wt% nanosilica made the geopolymer cement paste better in the face of failure, that the collapse did not occur suddenly. This is consistent with what was reported by [22,25], where the inclusion of nanosilica in the geopolymer cement paste gave rise to a residual stress phrase after failure. The cause of the emergence of residual stress is because the nanosilica was able to fill the pores and form a good matrix with epoxy. However, the nanosilica could not withstand the tensile load after cracking, so that the maximum tensile strength dropped. Therefore, the addition of nanosilica is recommended because it can increase the residual strength.

Figure 5 shows the crack pattern after the tensile test. The zigzag-shaped tensile pattern shows the material was trying to resist crack growth to achieve maximum strength. The fracture pattern that tends to be straight (Figure 5a) shows the material’s brittle nature, which was confirmed in the direct tensile strength test.

### 3.2. Ductility

Ductility is assessed from the relationship between load and displacement in the direct tensile strength test. In this study, ductility was tested by Equation (2), and the results are shown in Figure 6.

From Figure 6 it can be seen that geopolymer cement pastes with nanosilica concentration of 2 wt% have the highest ductility value of 2.08 compared to control specimens without using nanosilica. The geopolymer ductility of cement was 0.82, 2.08, 1.83, 1.34 for nanosilica concentrations of 0, 2, 3, and 4 wt%, respectively. The greater the ductility, the longer and sloping the post-yield phase. The addition of the nanosilica concentration by 2 wt% showed that the geopolymer paste was able to undergo deformation without experiencing sudden damage. The addition of nanosilica concentrations of 3 and 4 wt% did not show a significant increase in ductility.

It can be concluded that the addition of 2 wt% nanosilica made the geopolymer paste more ductile as indicated by the destruction that occurred unsuddenly. The ductility of the geopolymer paste with the addition of 2 wt% nanosilica was 2.52 times higher than that of the geopolymer paste without the addition of nanosilica. The increase in ductility was due to the addition of 2 wt% nanosilica, which inhibited propagation crack, and it took a longer time to disintegrate. The use of high ductility materials is recommended in earthquake-resistant buildings because the structural collapse does not occur suddenly.

### 3.3. Compressive Strength and Flexural Strength

The results of the compressive strength of the geopolymer cement paste are shown in Figure 7 and indicate the same trend with the flexural strength. The compressive strength values with the addition of 0, 2, 3, 4 wt% nanosilica were 21.47, 26.26, 25.50, and 18.16 MPa, respectively. Likewise, the average flexural strengths were 1.44, 2.62, 1.07, and 0.45 MPa, respectively.

Figure 7 shows an increase in compressive strength of 22% at the addition of 3 wt% nanosilica. Furthermore, the addition of 3 to 4 wt% nanosolica actually decreased the compressive strength by 3 and 31%, respectively. Furthermore, the flexural strength increased with the increase in the concentration of nanosica by 2 wt% but continued to decrease with the addition of 3 and 4 wt% nanosilica. Flexural strength increased by about 82% with 2 wt% nanosilica.

The result indicates that flexural strength follows the same trend as the compressive strength, and it was confirmed by other studies that the addition of nanoparticles increases the strength of the geopolymer cement paste [26,27]. The addition of 3 wt% nanosilica particles began to affect the compressive and flexural strength of the geopolymer. The compressive strength at 28 days of 3 and 4 wt% of the nanosilica was reduced to 25.50 and 18.16 MPa respectively, while the corresponding flexural strength fell to 1.07 and 0.45 MPa respectively.

The geopolymer paste’s compressive strength and flexural strength increased significantly due to the increase in the reaction product of the geopolymer matrix [26,28]. In the presence of a high alkaline environment and calcium from fly ash and additional nanosilica can react and form CSH or CASH and NASH gels and cause higher strength geopolymer pastes [29,30]. The results show that the geopolymer properties of high calcium fly ash could be improved by adding 2 wt% by weight of nanosilica. The increased flexural strength of the 2 wt% nanosilica was due to an improvement in the pore-filling mechanism. When the nanosilica was uniformly dispersed, it filled the voids in the matrix, created a denser microstructure [31,32]. This can be attributed to good interfacial adhesion leading to resistance to bending and fracture. Nanosilica of 2 wt% can bridge microcracks and increase crack growth resistance, leading to increased flexural strength.

The decrease in mechanical properties that occurred in the addition of 3 and 4 wt% nanosilica was due to the addition of nanosilica caused an increase in the porosity of the geopolymer paste, which contributed to the degradation of the mechanical characteristics. Another cause is the agglomeration of the nanosilica in the geopolymer matrix, that created microvoid, where is a weak zone in the material. Such agglomeration may result from poor dispersion of the nanosilica at higher concentrations [33,34]. In general, the nanosilica content in geopolymers has a major influence on the behavior of composites under bending.

Based on analysis of compressive strength and flexural strength, it can be concluded that 2 wt% nanosilica was the optimum value to increase the geopolymer reaction and act as a cavity-filler. The concentration addition of nanosilica by more than 2 wt% did not increase compressive and flexural strength. This was because the addition of nanosilica caused a poor dispersion between the nanosilica and the geopolymer matrix, and microvoid formation occurred, which weakens the geopolymer. On the other hand, there was a decrease in compressive and flexural strength with the addition of more than 2 wt% by weight of nanosilica due to the accumulation of nanosilica, which caused the formation of weak areas in the form of pores. After the curing process, these accumulated particles acted as stress concentrators in the cement-containing matrix and caused cracking which was also proved by SEM testing. Many studies have been conducted to determine the optimal nanocomposite content in geopolymers, and it was found that excessive doses can lead to a dramatic reduction in material characteristics [35,36].

The relationship between the compressive, flexural and tensile strength of geopolymer pastes is shown in Table 4. It shows that compressive strength was closely correlated with flexural strength; the higher the compressive strength, the higher the flexural strength. The addition of 2 wt% nanosilica resulted in a ratio of flexural to the compressive strength of 10%. Tensile strength and compressive strength are not well-correlated, and neither are the tensile and flexural strength.

### 3.4. Fracture Toughness

The fracture toughness of the geopolymer paste as a function of nanosilica content and controls is shown in Figure 8. Geopolymers containing nanosilica showed a significant increase in fracture toughness. Nanosilica is believed to increase fracture toughness, the energy from crack deflection at the particle and matrix interface, and debond the particles [37,38]. The addition of 2 wt% nanosilica to the geopolymer paste increased the fracture toughness, namely 0.6 MPa.m^1/2^ for geopolymer control and 1.06 MPa.m^1/2^ for nanocomposites. The increased fracture toughness of the nano geopolymer composites over the control geopolymer can be attributed to the good dispersion of the nanosilica throughout the matrix and the ability to resist crack propagation, and increase fracture toughness. Nanoparticles could improve the microstructure of geopolymer nanocomposites to enhance their mechanical properties due to their reactivity.

The results of this fracture toughness were confirmed by SEM observations, as shown in Figure 9a–h. It can be seen that the fracture surface of the control specimen shows a significant difference from the specimen with 2 wt% nanosilica (Figure 9c,d). A larger number of fly ash particles that partially reacted or did not react with pores or voids is shown in Figure 9a,b, whereas the dense microstructure was observed in the geopolymer containing 2 wt% nanosilica (Figure 9b,c). Nanosilica contributed to the geopolymer reaction and acted as a filler to improve the microstructure of the geopolymer with fewer pores and cracks. This showed that the nanosilica particles were able to improve the quality of the nanocomposite interface bonding, resulting in higher strength. In contrast, the addition of 3–4 wt% nanosilica resulted in many branched cracks and cavities on the surface (see Figure 9e,h). The matrix appeared less dense than the other mixtures, suggesting that the rate of addition of these nanosilica was excessive.

### 3.5. SEM Analysis

The SEM photo of the geopolymer paste is shown in Figure 9. The control paste contained a less dense matrix with more unreacted and partially reacted fly ash particles in the matrix (Figure 9a,b). For 2 wt% nanosilica (Figure 9c,d), fewer fly ash particles were observed, and the matrix appeared denser than the control paste. The use of nanosilica in geopolymers with high calcium fly ash resulted in the formation of additional CSH or CASH gel side by side with the NASH gel [39]. It was confirmed by the compressive strength that led to an increase in strength. On the other hand, the addition of nanosilica (Figure 9e,h) resulted in a large amount of observed nanosilica, and the matrix appeared to be less dense than the other mixtures. This suggests that the amount of addition of nanoparticles at a rate of 3–4 wt% was excessive.

Figure 10 shows the geopolymerization product in the form of C–S–H gel which occurred from a mixture of 2 wt% nanosilica, which in turn interacted with fly ash, as confirmed by the EDX results. Fly ash that is high in calcium, alumina, and silicates in the mixture caused the formation of calcium alumina–silicate hydrate (C–A–S–H) gel [40,41]. Therefore, fly ash removed additional amounts of silica from the nanosilica and contributed to the additional binder product, which influenced the regulation of geopolymer gel behavior [42]. From Figure 10, we can observe the emergence of Mg^2 +^ sourced from fly ash which contributed to the formation of a new gel phase as Na–Al (Mg) –Si–H gel, which was proven by EDX analysis. The micro-cracks observed in some SEM features may have occurred during mechanical testing because the powders tested for SEM were taken from the specimens after mechanical testing. Such micro-gaps could also result from internal stress occurring between particles during microstructural development [43].

Figure 11 illustrates a schematic model of geopolymer bonding as suggested by Davidovits [44]. It can be seen that the presence of magnesium in the geopolymer chain provides chemical stability or the so-called interatomic bonds in the matrix, due to the formation of different links such as Si–O–Mg, Si–O–Al, Ca–O–Si, and Si–O–Si. Therefore, it is believed that the intermolecular forces created in the system can increase mechanical strength.

### 3.6. XRD Analysis

The XRD patterns of fly ash-based geopolymers are shown in Figure 12. From this figure, it can be seen that the main crystal peak identified from the geopolymer paste is the quartz (SiO_2_) with high intensity at 2θ = 27°. Another crystal peak that emerged was Mullite (Al_6_O_13_Si_2_) where these crystals were detected in a range of 2θ (17° and 65°) [41]. Another pattern shown from the XRD test was the appearance of magnesium silicate (MgSiO_3_) crystals phases at 2θ = 32° and 37°. This phase arose due to the reaction of fly ash containing Mg with nanosilica in the geopolymer paste.

From XRD testing, it can be inferred that the raw material affected the formation of geopolymerization. The decrease in the intension of fly ash and geopolymer paste seen at 2θ = 27° was due to the influence of the alkaline solution [46]. Amorphous silica and aluminosilicates in fly ash and nanosilica in an alkaline environment are required for the formation of aluminosilicate gel which made the microstructure more homogeneous and increased the strength of the resulting geopolymer cement [47].

### 3.7. FTIR Analysis

The FTIR spectra of the geopolymer paste with the addition of nanosilica are shown in Figure 13. The band at 805 cm^−1^ is associated with the presence of stretching vibrations of Si–O and Al–O [46,48]. This is also confirmed by the fly ash base material which contains silica and aluminate. The difference in functional groups formed in the geopolymerization process is indicated by differences in absorption frequency. Bands at 940–1005 cm^−1^ represent asymmetric stretching of Si–O–Si and Al–O–Si vibrations formed during SiO_2_ dissolution and this indicates the appearance of C–S–H gel [49,50]. Vibration bands around 950–1200 cm^−1^ show the spectrum associated with T–O–Si (T: Si or Al) [51]. The bands at 1600 cm^−1^ present the H–O–H group [45]. A wide stretch band at 2500–3600 cm^−1^ indicates O–H stretching [52]. The bands at 2300–2400 cm^−1^ appeared due to a mixture of the activator solution and the binder during the geopolymerization process. This occurred because excess sodium in the alkaline solution concentration was transferred to the surface [53].

The conclusion can be drawn from the FTIR analysis that the geopolymer process occurred both in the specimens without the addition and with the addition of nanosilica. This is evidenced by the presence of the Si–O–Si and Al–O–Si functional groups. There were differences in the spectrum of the specimens with the addition of nanosilica and without the addition. This shows that the nanosilica is influential in forming other functional groups which increase the mechanical strength. Although the addition of 4 wt% nanosilica decreased the compressive strength, this could not be confirmed clearly by FTIR.

## 4. Conclusions

The mechanical and microstructural properties of physically processed nanosilica-based geopolymer pastes have been investigated experimentally in this study. According to the results, nanosilica influences mechanical strength, fracture toughness, and microstructure. It was found that the tensile strength showed a reduction after nanosilica addition where the tensile strength decreased by 31%. However, compressive strength, flexural strength, and fracture toughness were improved in the presence of nanosilica, confirming that they can be used as a reinforcing material.

The compressive strength increased by up to 22% at 2 wt% of nanosilica and continued to decrease with the addition of 3 and 4 wt% nanosilica. The same trend was confirmed in flexural strength and fracture toughness which each increased by 82%. The addition of 2 wt% of nanosilica made the material more ductile by 2.52 times. This indicates that nanosilica is suitable for use as a material in geopolymer cement because of its ability to inhibit the propagation crack and the possibility of structural collapse that occurred unsuddenly.

Microscopy images of the geopolymer cement paste showed good nanosilica dispersion in the proportion of 2 wt% with less pore and matrix density. EDX analysis showed the emergence of phase gels C–S–H, A–S–H, and Na–Al (Mg) –Si–H which leads to an increase in the mechanical properties of the geopolymer product. XRD results showed that nanosilica affected the formation of geopolymerization. Aluminosilicate gel made the microstructure more homogeneous and increased the strength of the geopolymer cement paste. We conclude that nanosilica from rice husk ash is suitable for use as an alternative source of cementitious material in cementitious geopolymers.

## Figures and Tables

**Figure 1 polymers-13-02178-f001:**
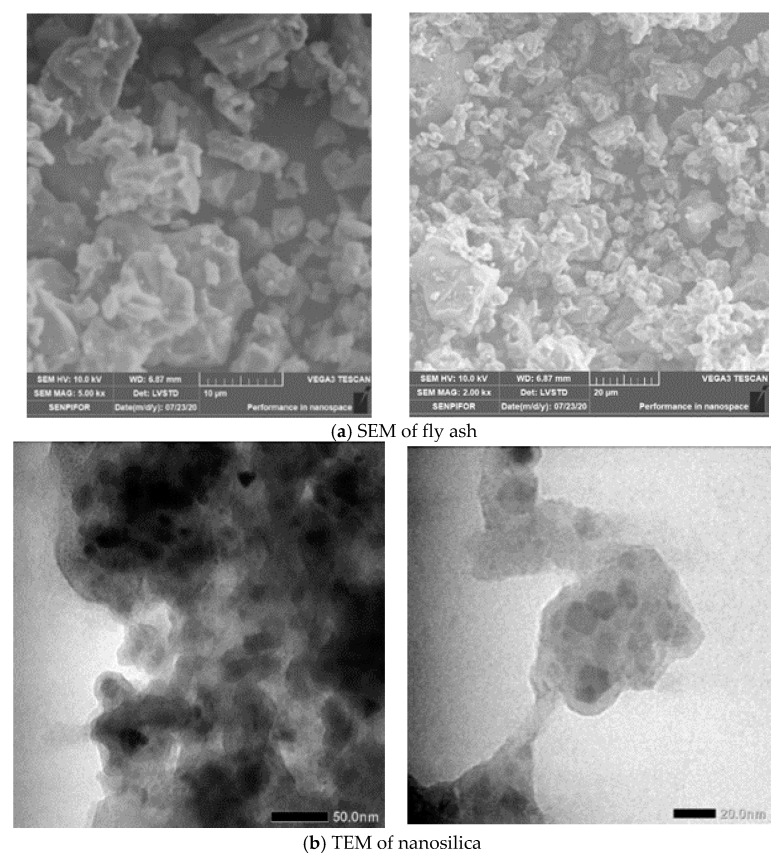
The morphology of raw materials: (**a**) fly ash and (**b**) nanosilica.

**Figure 2 polymers-13-02178-f002:**
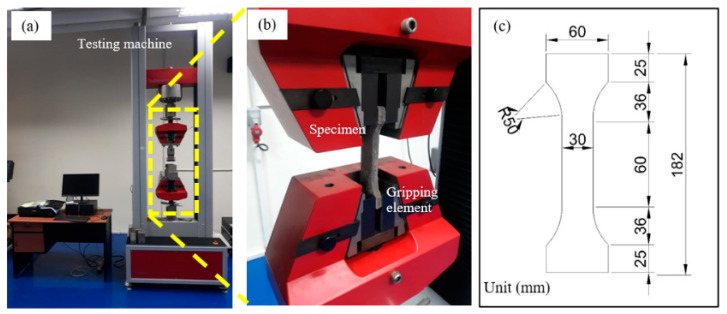
(**a**) Testing machine, (**b**) specimen in gripping element, (**c**) the dimension of typical dog bone specimen.

**Figure 3 polymers-13-02178-f003:**
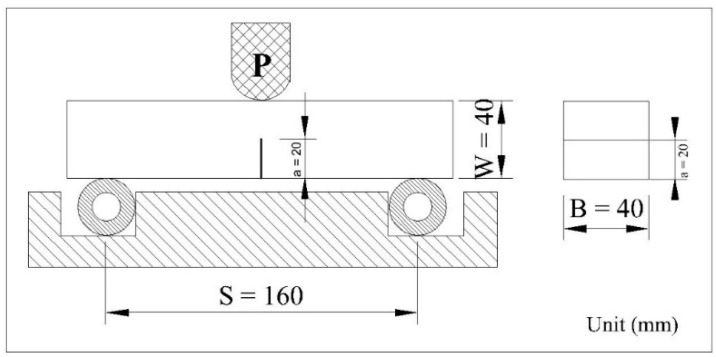
Schematic of three-point bending test for the fracture toughness tests.

**Figure 4 polymers-13-02178-f004:**
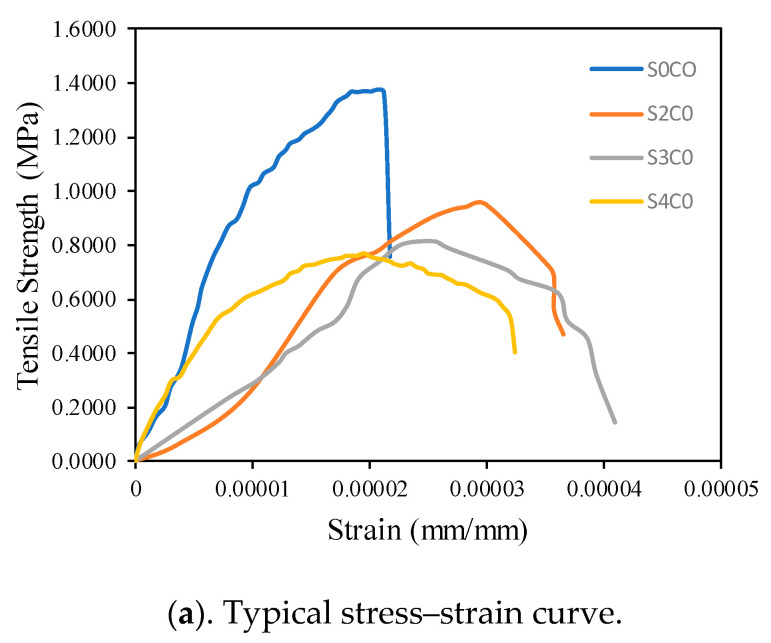
Direct tensile tests results: (**a**) Typical stress–strain curve, (**b**) maximum tensile strength.

**Figure 5 polymers-13-02178-f005:**
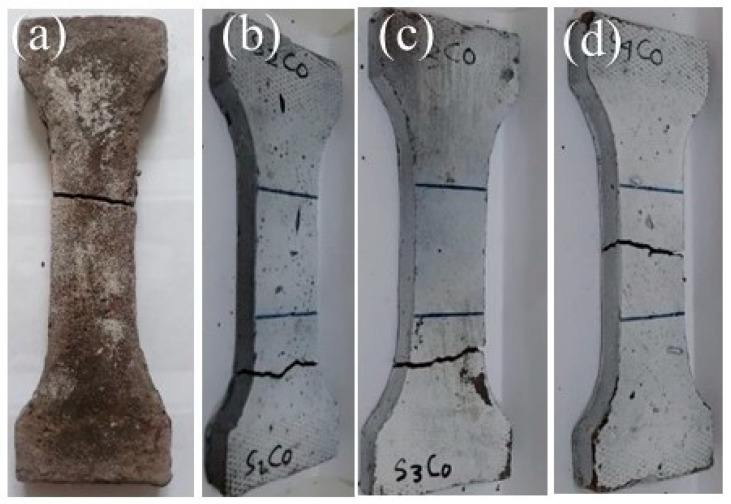
Crack condition after direct tensile strength test (**a**) 0 wt% nanosilica, (**b**) 2 wt% nanosilica, (**c**) 3 wt% nanosilica, (**d**) 4 wt% nanosilica.

**Figure 6 polymers-13-02178-f006:**
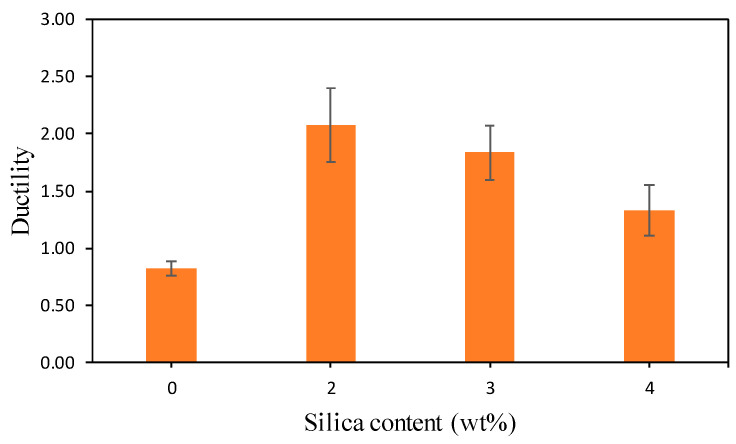
Ductility geopolymer cement with nanosilica.

**Figure 7 polymers-13-02178-f007:**
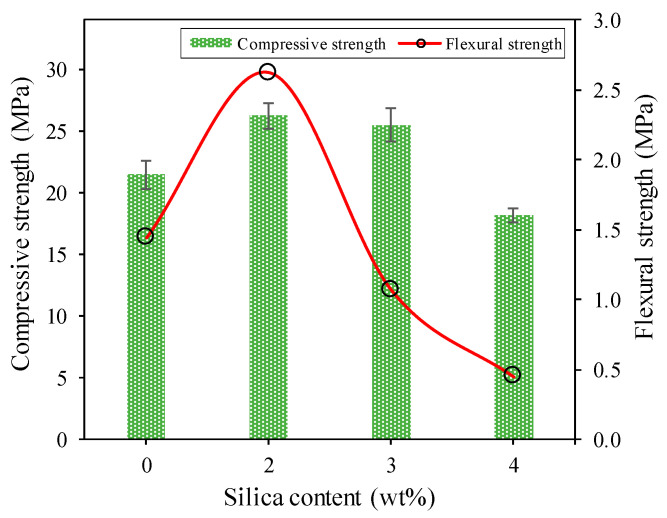
Compressive strength and flexural strength of geopolymer with some silica content.

**Figure 8 polymers-13-02178-f008:**
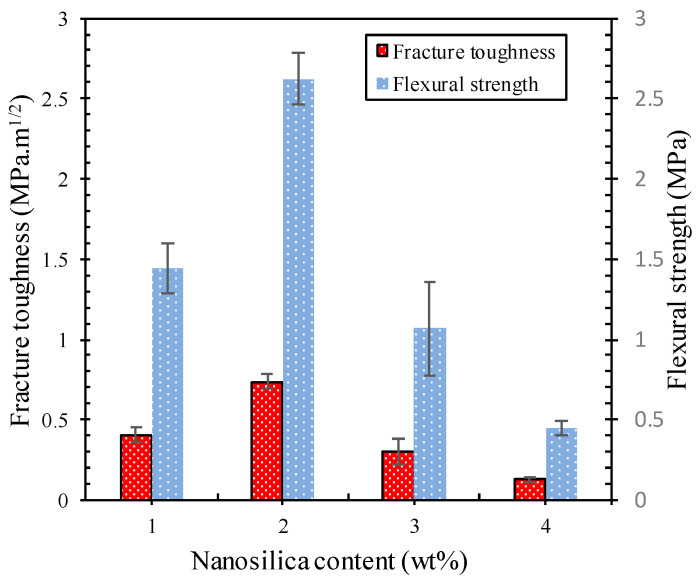
Fracture toughness and flexural strength of geopolymer pasta as a function of nanosilica content.

**Figure 9 polymers-13-02178-f009:**
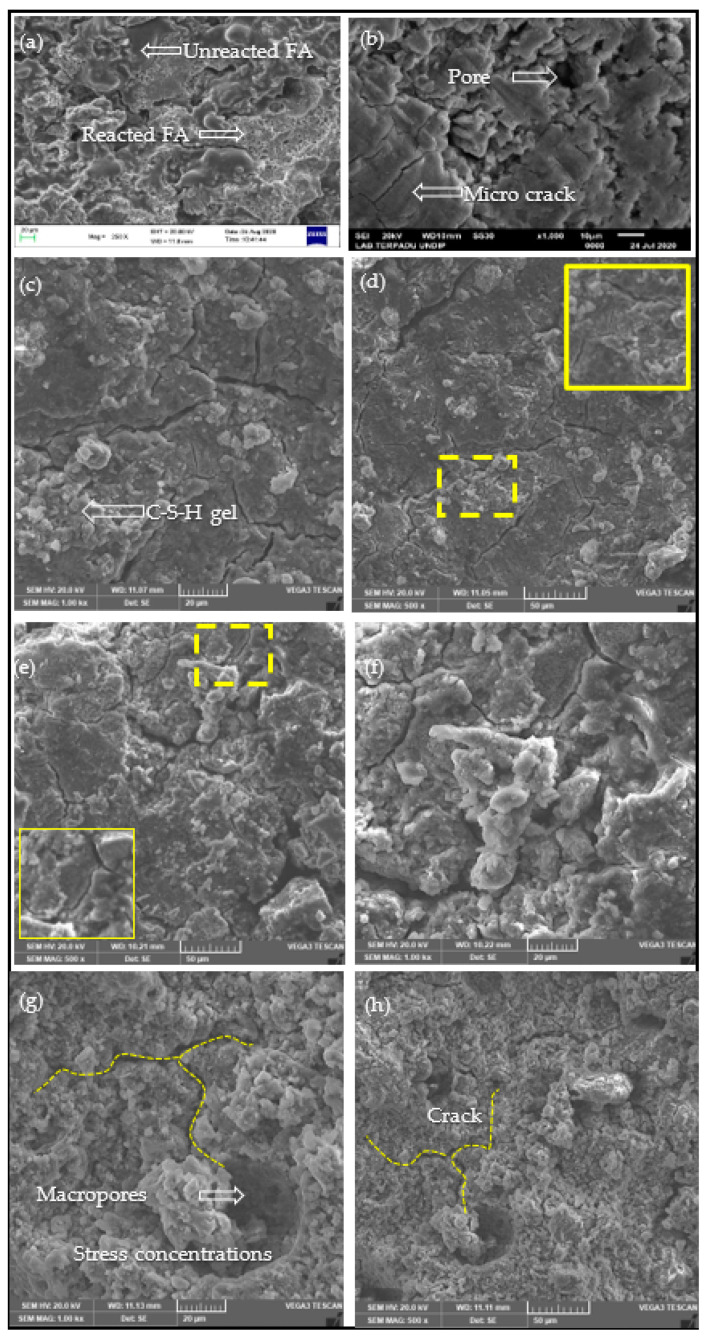
SEM images of geopolymer paste with various nanosilica content: (**a**,**b**) 0 wt%, (**c**,**d**) 2 wt%, (**e**,**f**) 3 wt%, (**g**,**h**) 4 wt%.

**Figure 10 polymers-13-02178-f010:**
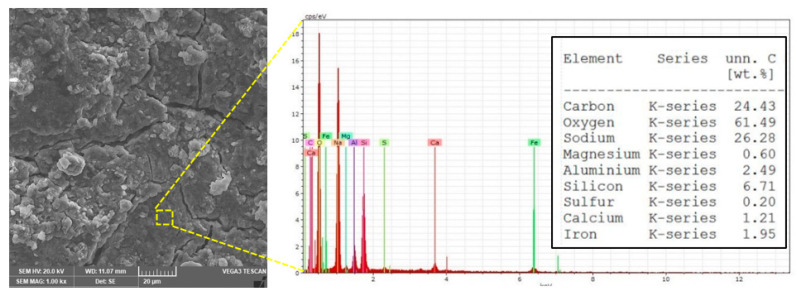
SEM/EDX of geopolymer paste.

**Figure 11 polymers-13-02178-f011:**
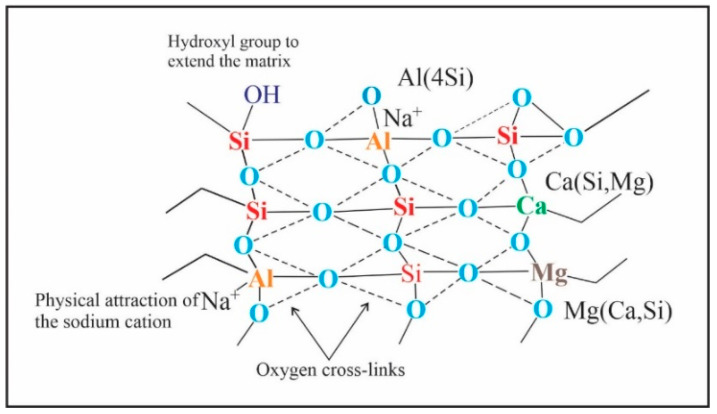
Schematic model of geopolymer bonding [45], with permission from Elsevier, 2021.

**Figure 12 polymers-13-02178-f012:**
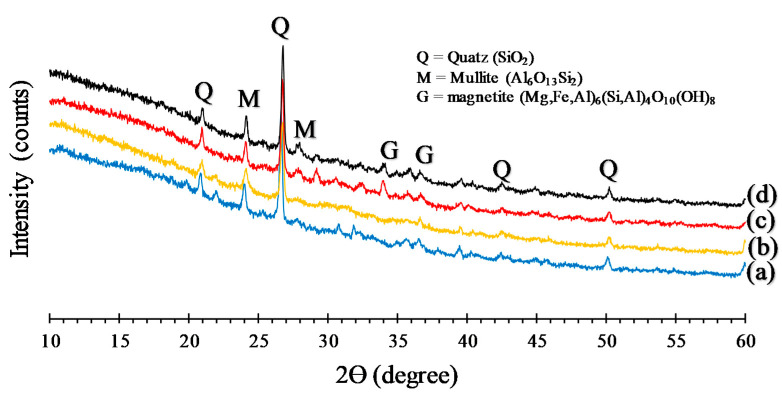
X-ray diffraction patterns of geopolymer paste: (**a**) 0 wt% nanosilica, (**b**) 2 wt% nanosilica, (**c**) 3 wt% nanosilica, (**d**) 4 wt% nanosilica.

**Figure 13 polymers-13-02178-f013:**
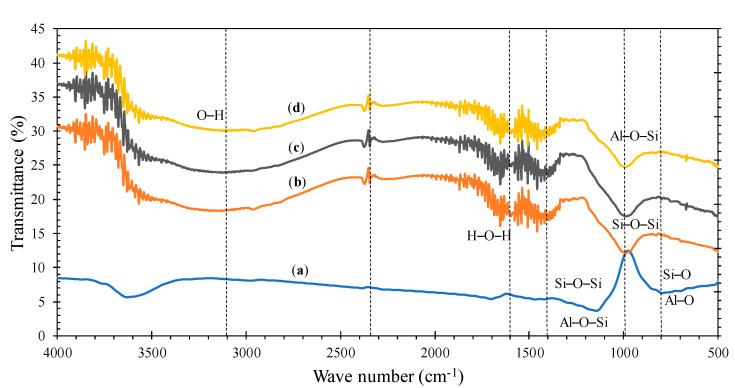
FTIR spectra of the geopolymer paste with various addition of nanosilica, (**a**) 0 wt%, (**b**) 2 wt%, (**c**) 3 wt%, (**d**) 4 wt%.

**Table 1 polymers-13-02178-t001:** Chemical composition of fly ash and nanosilica.

Chemical Analysis	Class C Fly Ash ^a^ (wt%)	Nanosilica (wt%)
SiO_2_	21.07	91.78
Al_2_O_3_	9.65	-
Fe_2_O_3_	27.23	0.15
CaO	32.58	2.09
MnO	0.44	0.18
K_2_O	1.17	4.11
SO_3_	5.69	1.43
TiO_2_	1.68	-
Cl	0.22	-
Ag_2_O	0.23	-
Yb_2_O_3_	0.09	-
P_2_O_5_	-	0.13
SiO	-	0.13

^a^ ASTM C 618.

**Table 2 polymers-13-02178-t002:** Thermophysical properties of materials.

Thermophysical Properties	Materials/Value	
	Fly Ash	Nanosilica
Size average	30 µm	339.09 nm
Density	2.45 gr/cm^3^	3.41 g/cc
Surface area	-	28.566 m²/g
Pore Radius	-	1.320 × 10 Å
Temperature onsite	57.04 °C	736.09 °C
Weight loss 600 °C	6.58%	1.3%
Material properties from XRD	Crystallinity (64.9%), amorf (35.09%)	semi-crystalline

**Table 3 polymers-13-02178-t003:** Mixture proportions of geopolymers paste.

Specimens	Fly Ash	NaOH Solution	Na_2_SiO_3_ Solution	Epoxy Resin	Water	Nanosilica	Nanosilica Content
	(g)	(g)	(g)	(g)	(mL)	(g)	(wt%)
1	1000	260	260	434	650	0	0
2	980	260	260	434	650	20	2
3	970	260	260	434	650	30	3
4	960	260	260	434	650	40	4

**Table 4 polymers-13-02178-t004:** The relation between compressive, flexural, and tensile strength of geopolymer paste.

Nanosilica Content (wt%)	Strength of Geopolymer Paste (MPa)			Ratio (%)		
	Compressive	Flexural	Tensile	Flexural to Compressive Strength	Tensile Strength to Compressive Strength	Tensile to Flexural Strength
0	21.47 ± 1.12	1.44 ± 0.16	1.37 ± 0.01	7	6	95
2	26.26 ± 1.03	2.62 ± 0.16	0.95 ± 0.02	10	4	36
3	25.50 ± 1.33	1.07 ± 0.29	0.82 ± 0.06	4	3	76
4	18.16 ± 0.56	0.45 ± 0.04	0.77 ± 0.02	2	4	171

## Data Availability

The data presented in this study are available on request from the corresponding author.

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
