# Peer review of "The Effects of Nanosilica on Mechanical Properties and Fracture Toughness of Geopolymer Cement"

_polymers, 2021, doi:10.3390/polym13132178_

Round 1
Reviewer 1 Report
General comments: This study aimed to investigate the effect of physically processed nanosilica on geopolymer cement's mechanical/microstructure properties. The mechanical strength and microstructural development of geopolymer paste samples containing different nanosilica are treated in the curing process at room temperature studied.
Specific Issues:
- Please eliminate multiple references. After that, please check the manuscript thoroughly and eliminate ALL the lumps in the manuscript. This should be done by characterising each reference individually and by mentioning 1 or 2 phrases per reference to show how it is different from the others and why it deserves mentioning. Multiple references are of no use for a reader and can substitute even a kind of plagiarism, as sometimes authors are using them without proper studies of all references used. In the case, each reference should be justified by it is used and at least short assessment provided.
-
How Authors calculate standard deviation for compressive strengths loss and weight changes?
- Thermophysical properties of material should be tabled.
-
How Authors calculate the size distribution of nanoparticles ? Explain in the methodology section also.
-
What is the average depth explored in Authors samples through the following characterization techniques: XRD, XPS ?
-
What is the relationship of compressive, flexural and tensile strength of sample?
-
How does the Author interpret EDS results showing %wt of different elements? Does this %wt tell about the concentration of the elements or what?
-
How to Author address family-wise (Type I) errors in SEM analysis?
Author Response
Thank you for the encouraging and valuable comments. Please see the attachment

Reviewer 2 Report
The manuscript is an interesting contribution in the preparation and characterization of fly ash geopolymers. The authors carried out a good experimental section to characterize mechanically e spectroscopically the prepared samples. I would suggest some minor changes, see the attached report, before publication.

Author Response
Thank you for the encouraging and valuable comments. We have corrected the manusript according to your suggestions and We have marked the parts that we have corrected. Please see the attachment

Round 2
Reviewer 1 Report
All comments are addressed
Reviewer 2 Report
The authors revised their manuscript according to the reviewer suggestion